behaviour/environmental science/complexity

evolutionary game theory, social dilemma, cooperation, conservation, sustainability

**Author for correspondence:**
C. Gracia-Lázaro
e-mail: cgracia@bifi.es

†These authors contributed equally to this study.

# Behavioural patterns behind the demise of the commons across different cultures

M. Jusup[1,†], F. Maciel-Cardoso[2,†], C. Gracia-Lázaro[2], C. Liu[3], Z. Wang[4] and Y. Moreno[2,5,6]

[1]Tokyo Tech World Research Hub Initiative (WRHI), Institute of Innovative Research, Tokyo Institute of Technology, Tokyo 152-8552, Japan
[2]Institute for Biocomputation and Physics of Complex Systems, University of Zaragoza 50018, Spain
[3]Center for Ecology and Environmental Sciences, and [4]Center for OPTical IMagery Analysis and Learning, Northwestern Polytechnical University, Xi'an 710072, People's Republic of China
[5]Department of Theoretical Physics, Faculty of Sciences, University of Zaragoza 50009, Spain
[6]ISI Foundation, Turin 10126, Italy

MJ, 0000-0002-0777-0425; FM-C, 0000-0002-2789-9250; CG-L, 0000-0002-9769-8796; ZW, 0000-0002-8182-2852; YM, 0000-0002-0895-1893

Common-pool resources require a dose of self-restraint to ensure sustainable exploitation, but this has often proven elusive in practice. To understand why, and characterize behaviours towards ecological systems in general, we devised a social dilemma experiment in which participants gain profit from harvesting a virtual forest vulnerable to overexploitation. Out of 16 Chinese and 15 Spanish player groups, only one group from each country converged to the forest's maximum sustainable yield. All other groups were overzealous, with about half of them surpassing or on the way to surpass a no-recovery threshold. Computational–statistical analyses attribute such outcomes to an interplay between three prominent player behaviours, two of which are subject to decision-making 'inertia' that causes near blindness to the resource state. These behaviours, being equally pervasive among players from both nations, imply that the commons fall victim to behavioural patterns robust to confounding factors such as age, education and culture.

## 1. Introduction

Instances of overused common-pool resources abound in human history. Among the more famous are the crash of the Peruvian anchoveta fishery in the early 1970s [1] and the overfishing-induced ecosystem regime shift off the coast of Newfoundland in the early 1990s [2]. That these are but symptoms of a global trend is documented by the Food and Agriculture Organization of the

United Nations whose data show that about one in four of the world's fisheries collapsed between 1950 and 2000 [3]. Blatant overexploitation extends beyond fish too. Protected Bornean rainforests in Kalimantan lost over 56% of their geographical span between 1985 and 2001, much of it due to unsanctioned logging [4]. In fact, over 10% of worldwide timber trade is illegal, amounting to a staggering $15 billion annually based on estimates from the early 2000s [5]. All these are classic examples of 'the tragedy of the commons', a term coined by Garrett Hardin in 1968 [6,7] to describe generally poor human stewardship of nature's riches.

Access to common-pool resources is by definition difficult to control [8,9], implying a need for user self-restraint to ensure sustainability. If users, e.g. fishers or loggers, are free from surveillance, they face a dilemma to either exploit the resource sustainably as a form of cooperation or overuse the resource for immediate profit as a form of defection. Although this social dilemma is not without successful resolutions [10,11], there are no panaceas either [12].

Systematic attempts to resolve social dilemmas are in the domain of (evolutionary) game theory [13,14], within which the governance of the commons as a collective action problem [8] has often been studied in the form of public goods games [15–17] or more specialized common-pool resource games [18–21]. Early experiments concluded that behaviour is often closer to free riding than the Pareto efficiency [22]. Selfishness, however, is not necessarily a chief human impulse [23] as evidenced by various experimental treatments that entice cooperation [24–26]. Common-pool resources, in particular, have proven manageable even in the absence of an external authority if communication channels between participants are made available [21,27,28] or rent dissipation from harmful competition can be reduced with a proper rights-based management protocol [29].

Given that full information on the underlying dynamics of a common-pool resource is usually unavailable, are human decision makers able to identify the optimal level of exploitation? We incorporated resource dynamics into an experimental platform to gain empirical insights into decision making when exploiting common-pool resources (see Material and methods and electronic supplementary material, Methods for details). Our purpose was to put participants in a situation that closely mimics epistemic and socio-economic realities of resource exploitation [30,31]. Specifically, when dealing with biological resources, e.g. a fish stock or a forest, a broad outline of the resource's dynamics is typically knowable, yet many key details, such as the point of optimal population growth or the population's tipping points, remain hidden under the veil of complexity [32–34]. From the socio-economic perspective, exploitation is most often done for profit, with comparisons in terms of various profitability indicators being of utmost importance to business owners. Our experiment with human participants, described hereafter, not only demonstrates that the demise of the commons is a serious threat in these conditions, but also pinpoints a unifying cause behind robust behavioural patterns displayed by two geo-socially distant populations.

Participants in the experiment engaged in a social-dilemma game (figure 1) in which groups of six individuals exploited a virtual forest over the course of 50 rounds, each round representing a year. The exact number of rounds, as participants were made aware, was undisclosed to avoid the final-round effects, i.e. a change in behaviour due to the impending end of the game. Each round consisted of inputting weekly logging effort (range 0–7 with decimal numbers permitted). Input efforts were fed to an ecological model to calculate the relative effects of tree regrowth and logging on the forest's state (electronic supplementary material, Note 1). Starting with 400 trees, the forest's regrowth rate was such that keeping the number of trees at around 151 would have produced the maximum sustainable yield (MSY). We blocked regrowth below a no-recovery threshold of 100 trees to emulate an Allee effect [35]. Although participants lacked information on the underlying ecological model, including the existence of MSY and the exact value of the no-recovery threshold, the forest's state could be monitored at all times via a detailed interface (electronic supplementary material, figure S1). Participants could also compare their own performance in terms of effort, yield and profit with others. We assumed effort to be costly, meaning that excessive logging in a heavily exploited forest could have generated losses. Using this set-up, we gathered a rich dataset of 9300 decisions from 96 undergraduate students in Xi'an, China and 90 individuals from the general population in Zaragoza, Spain (electronic supplementary material, table S1). We instructed participants that they would receive a monetary payout tied to their end profit in the game. The resulting payouts amounted to an average of ¥62.6 in China and €15.1 in Spain.

## 2. Results

The number of trees in the virtual forest corresponding to MSY (≈151 trees) gave us a natural performance classifier for 16 Chinese and 15 Spanish player groups. Namely, we defined the optimal

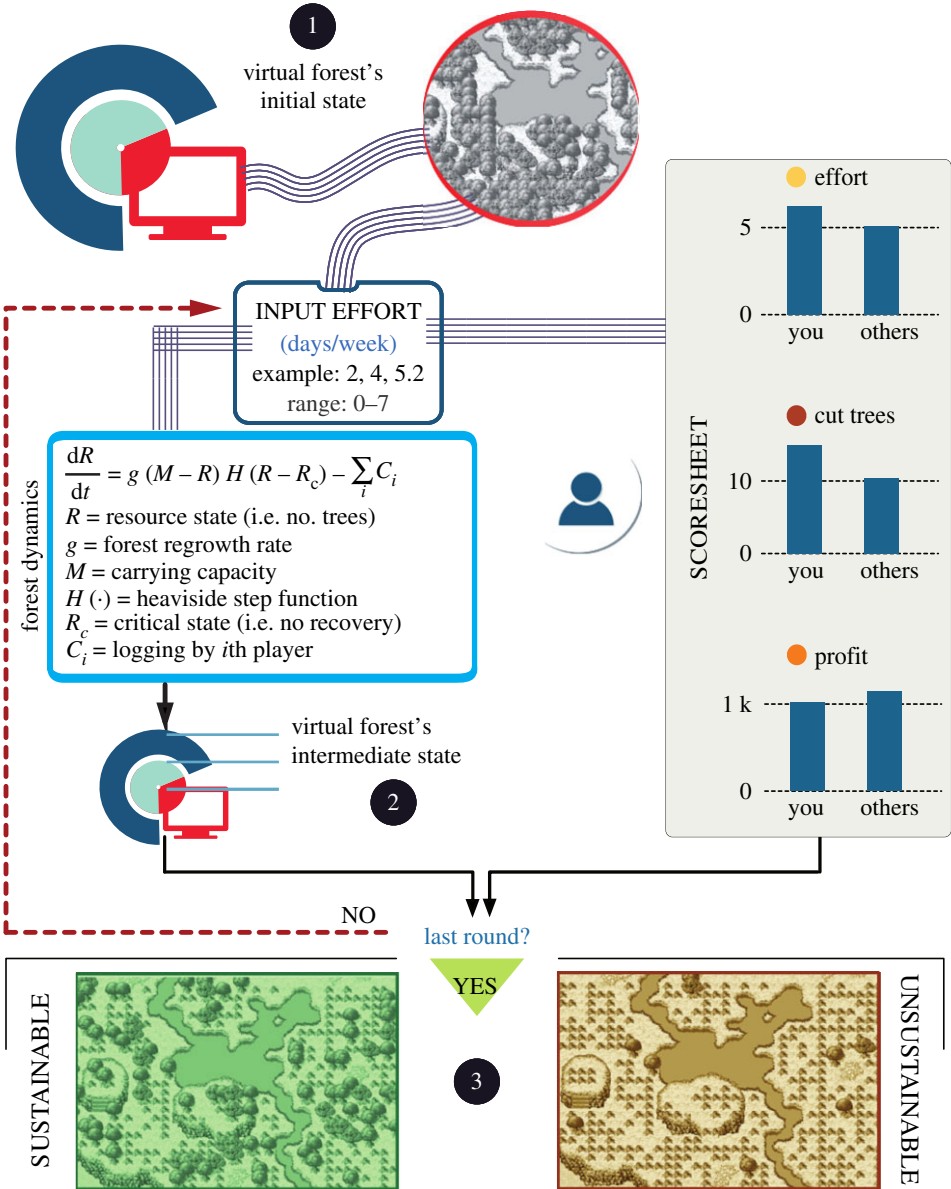

**Figure 1.** Game experiment with feedback between user inputs and resource dynamics. In step ① of the game, we displayed the virtual forest's initial state to participants and asked them to input their preferred weekly effort (range 0–7 with decimal numbers permitted). The system used these inputs in two ways. Participants could inspect a scoresheet to compare their own performance in terms of effort, yield and profit with others. Simultaneously, a mathematical model calculated the forest's new state by balancing regrowth and logging. In step ② of the game, participants used their scoresheets and the new forest state to decide on further effort. This step kept repeating until reaching a total of 50 rounds. In step ③, the platform presented the final outcome to participants, revealing whether they exploited the forest sustainably or not.

exploitation as any number of trees left for cutting after 50 rounds that was within ±10% of the MSY number (136–166 trees). Only one group from each of the countries was able to optimally exploit the resource. None of the groups underexploited the resource by ending the experiment above the optimal range, whereas a total of 14 Spanish and 15 Chinese groups overexploited the resource by ending below this range.

A cursory comparison of the virtual forest's time evolution for 14 Spanish and 15 Chinese groups who overexploited the resource suggests that the former groups performed much worse (figure 2a). Seven groups from Spain drove the number of trees in the forest below the no-recovery threshold (=100 trees), whereas only one group from China did the same, and it did so only in the last round of the game. A closer inspection of the data, however, reveals that multiple Chinese groups kept depleting the resource, and given more time, would have probably crossed the no-recovery threshold

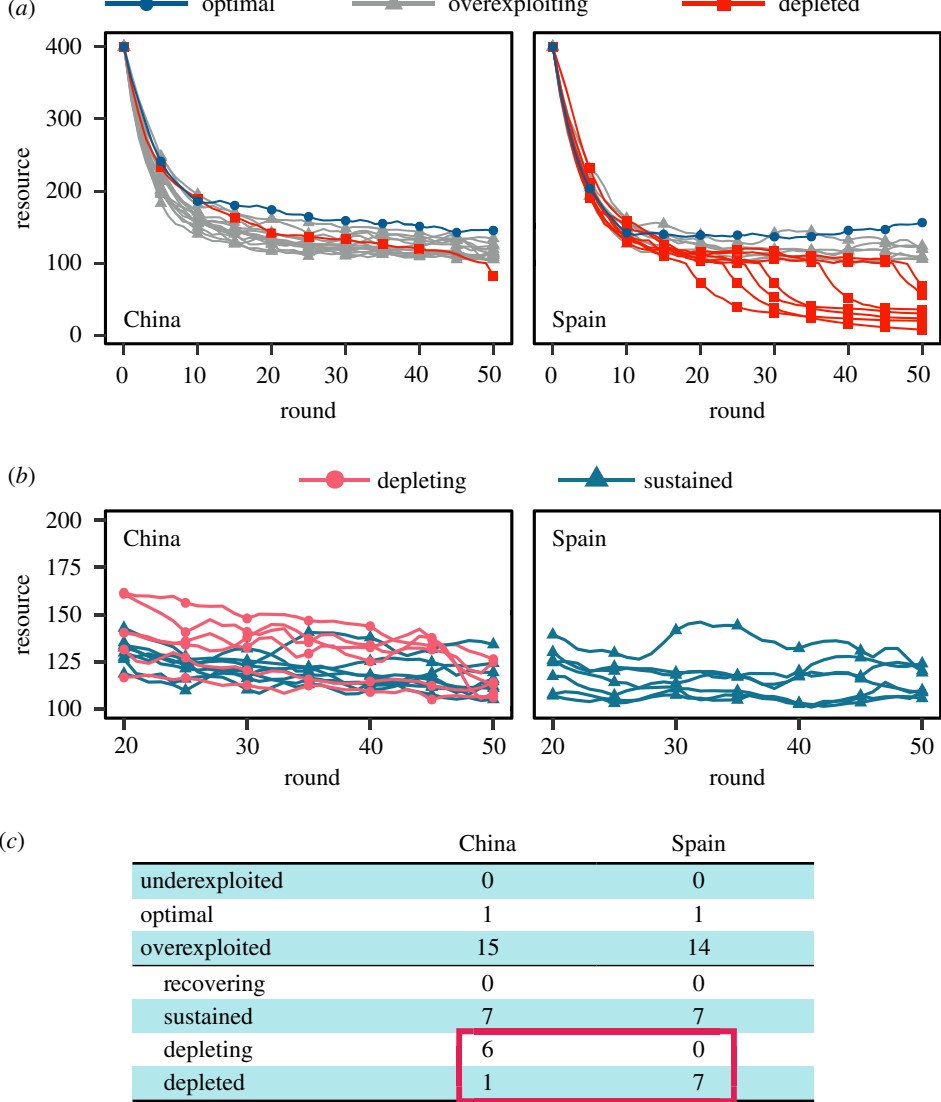

**Figure 2.** Overexploitation is irresistible. (*a*) Out of 16 Chinese and 15 Spanish groups who exploited a common-pool resource (i.e. a virtual forest), only one group from each country was able to keep the resource at an optimum. We defined the optimum as ±10% from the number of trees maximizing the sustainable yield (≈151 trees). Worryingly, all other groups overused the resource, and what is more, one Chinese and seven Spanish groups depleted it below the no-recovery threshold (=100 trees). (*b*) Chinese groups seemingly do better than their Spanish counterparts, but is this truly so? To examine the likely fate of overexploited, but non-depleted virtual forests beyond round 50, we tested whether after a transitory period of about 20 rounds the number of trees was recovering, sustained or depleting (electronic supplementary material, table S2). We found that six Chinese groups kept depleting the resource until the very end, while seven groups from each country sustained a relatively constant number of trees. No groups from either country managed to overturn the negative trend and allow the resource to recover. (*c*) 'The tragedy of the commons' perfectly summarizes the results of our experiment in China and Spain. Notably, none of the groups from the two countries underexploited the resource, while a total of seven groups from each country depleted or would have probably ended up depleting the resource (red rectangle).

too (figure 2*b*). Testing for the statistical significance of the negative trend in these data shows that at least six additional Chinese groups, but none of the Spanish groups, were in danger of crashing the resource (electronic supplementary material, table S2). A total of seven groups on each side sustained the overexploited resource, i.e. their exerted effort was sustainable, but they kept earning suboptimal profits (figure 2*c*). Interestingly, none of the overexploited resources have shown a recovering trend for the duration of the game. A conclusion is that the outcome in both countries, especially when considering together the groups who kept depleting or already depleted the resource (red rectangle in figure 2*c*), was remarkably similar and rather dismal.

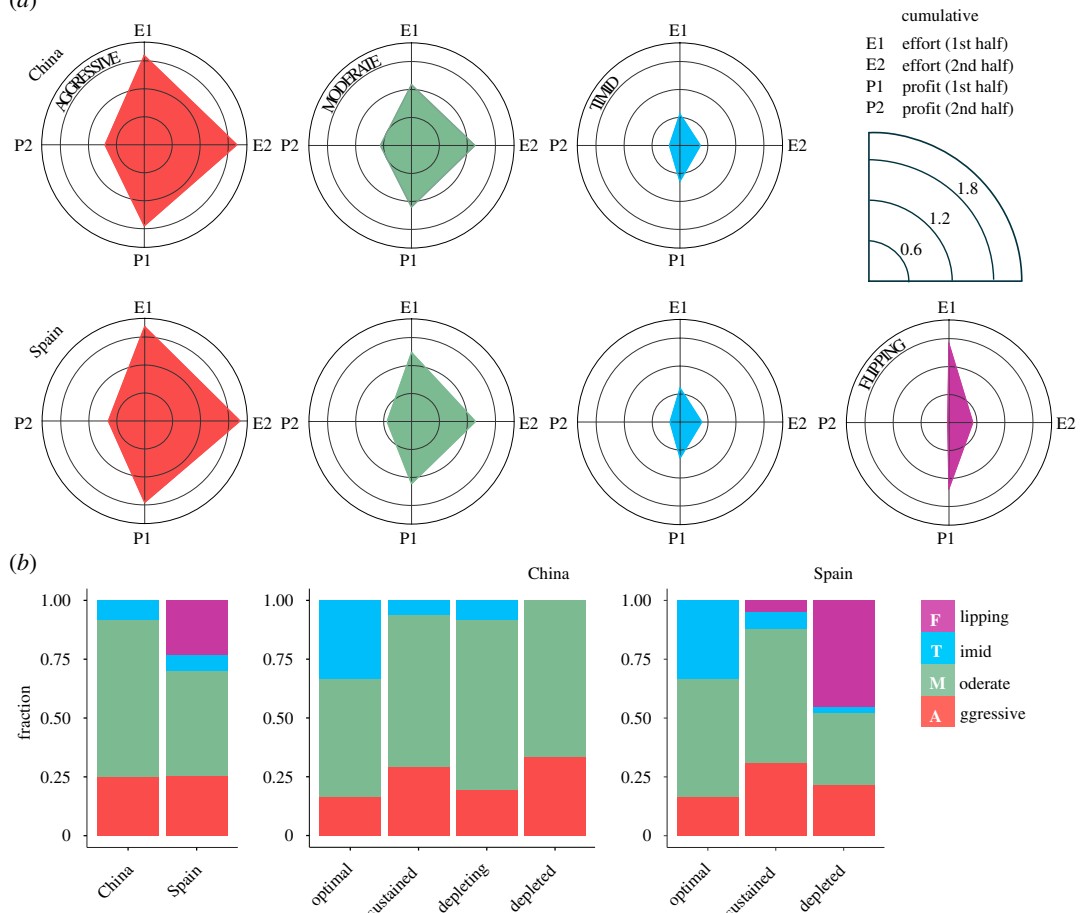

**Figure 3.** Interplay of prominent behaviours explains overexploitation. (*a*) Running a clustering algorithm on data from China and Spain separately, we identified three (respectively, four) distinct prominent behaviours among the Chinese (respectively, Spanish) participants. Apart from the behaviour unique to Spain, the three remaining behaviours are nearly identical irrespective of the country. In terms of effort, these can be described as aggressive, moderate and timid. With the scale set relative to the MSY effort and the corresponding profit, we see that aggressive players exceed the MSY effort by over 80%, earning large profits in the first half of the game. Moderates stay closer to the MSY effort, nonetheless exceeding it by about 20%. Timid players start cautiously at 60% of the MSY effort and, unlike aggressive or moderate players, reduce effort in response to resource deterioration. The fourth Spanish behaviour flips from an aggressive initial stance to a timid subsequent one (see also electronic supplementary material, figure S4*b,d*), earning almost no profit late in the game. (*b*) Overall abundance of aggressive and timid players is remarkably similar across countries (left panel), as is the abundance of these players in groups that played optimally and groups that sustained the resource in an overexploited state (middle and right panels). Optimal play clearly requires a much more favourable aggressive-to-timid ratio than is present in the overall abundance, thus explaining overexploitation. The flipping behaviour is nearly exclusive to groups that depleted the resource in Spain, indicating that many players become responsive to the resource state only when it is too late.

For a first glimpse into the underlying causes of similarly dismal outcomes in both countries, we resorted to the *k*-means clustering algorithm. The purpose was to identify prominent player behaviours based on four quantitative characteristics: cumulative efforts and total profits from the first and the second half of the game taken separately. The Chinese participants exhibit three prominent behaviours broadly describable as aggressive, moderate and timid (figure 3*a*). Effort and profit gradually decrease from aggressive to moderate to timid players. Remarkably, performing independent clustering on the Spanish data reveals exactly the same behaviours, with the addition of a fourth one, dubbed flipping (figure 3*a*). This last behaviour is aggressive or moderate in the first half of the game, but turns timid in the second half (electronic supplementary material, Note 2). We furthermore found that aggressive and timid behaviours are almost equally abundant in both countries, encompassing approximately 25% and approximately 10% of players, respectively (figure 3*a*). The Chinese case is enough to demonstrate that with such a distribution of players overexploitation is the most likely outcome. Adding the rather

aggressive first-half behaviour of flipping players to this only contributes to the faster resource decline in Spain than in China, thus helping to explain why multiple Spanish groups managed to even cross the no-recovery threshold.

Prominent player behaviours show what separates optimal harvesting from sustained overexploitation from resource depletion. Groups who harvest optimally have almost the same composition in both countries (figure 3b), characterized by a relative scarcity of aggressive (approx. 17%) and a disproportional abundance of timid (approx. 33%) players. Groups responsible for sustained overexploitation also have almost the same composition in both countries (figure 3b), only here aggressive players are abundant (approx. 30%) and timid players are scarce (approx. 7%). The Chinese group who depleted the resource has the highest proportion of aggressive players (approx. 33%) and no timid ones whatsoever (figure 3b), while the corresponding Spanish groups have only a few stray timid players (approx. 2.5%). The latter groups also harbour almost all flipping players (approx. 45%), who act rather aggressively in the first half of the game (electronic supplementary material, Note 2) and contribute to resource decline alongside aggressive players (approx. 21%). The four identified prominent behaviours thus go a long way in explaining the subtle differences in the virtual forest's time evolution between China and Spain, as well as the overall bias towards overexploitation. Particularly intriguing is a number of remarkable similarities between the two countries hinting at the existence of robust behavioural patterns behind the demise of the commons.

To gain an even deeper insight into the underlying causes of similarly dismal outcomes in both countries, we constructed a statistical regression model of participant behaviour. The model's dependent (i.e. response) variable was effort, which we tried to explain using several independent (i.e. explanatory) variables: the virtual forest's state as a primary external driver, up to five own lagged efforts to account for autocorrelations, and one lagged average effort of others to account for cross-correlations. These explanatory variables zoom in on the collective behaviour in the sense that the values of the accompanying parameters remain the same for all participants from a given country. Individual differences entered the model by allowing constant terms and residual variances to be participant-specific, thus respectively quantifying individualistic propensities to exert and randomly vary effort. The model is able to explain the posted efforts. That predictions fit observations well is seen in observation-versus-prediction scatter plots (electronic supplementary material, figure S5), on which points gather around the 'diagonal', i.e. the line with intercept 0 and slope 1. The coefficients of determination further indicate that the model accounts for nearly 60% (respectively, 70%) of the total variance in the Chinese (respectively, Spanish) data.

The behavioural regression model offers additional plausible reasons why outcomes in China and Spain were similarly dismal. We found among the Spanish participants that, while the virtual forest's state and the average effort of others inform decisions on the current effort, a key determinant in this context is one's own lagged efforts (figure 4a). We thus witnessed a form of decision-making 'inertia' by which past choices heavily weigh on the present choice. The effect is significant up to five lags in the past. Interestingly, the Chinese participants exhibit qualitatively the same behavioural patterns; again the forest's state and the average effort of others inform decisions, but these are much less influential than one's own lagged efforts (figure 4b). Even quantitatively the results are remarkably similar because only the effect of own effort at lag 2 is slightly weaker among the Chinese participants, while the effect at other lags is statistically indistinguishable between the two countries (figure 4b). The same is true for the average effort of others. Based on these results, two conclusions force themselves upon us. First, making decisions based on one's own past efforts rather than the resource state is akin to driving partly blind and staying on track because it has worked so far. Crashes are bound to happen. Second, given that a decent number of participants from vastly different countries performed in a remarkably similar manner, behavioural patterns behind the demise of the commons are, if not universal, then at least robust to a myriad of confounding factors.

The one substantial difference between the two countries is that the virtual forest's state correlates negatively with the effort of the Chinese, but positively with the effort of the Spanish participants (figure 4). The former start exploiting the resource more cautiously, but then compensate for a steady resource degradation with more effort. The latter, by contrast, start more aggressively, but then curtail their zeal in response to a disappearing resource. The described difference between the two countries helps to explain the faster resource depletion in Spain than in China (figure 2), and is fully consistent with the clustering results (figure 3). Analysing the participant-specific model terms further complements this explanation (electronic supplementary material, Note 3). Meticulous regression diagnostics show that we avoided the common pitfalls of this type of analysis, and thus that the model's results are credible (electronic supplementary material, Note 4).

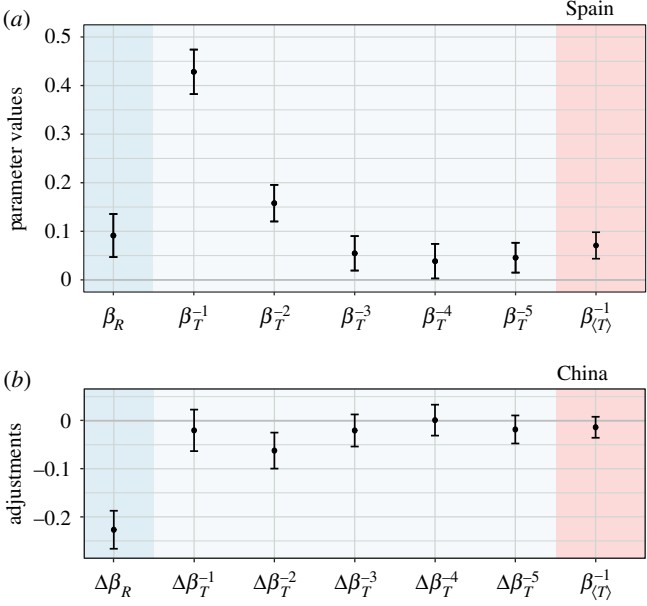

**Figure 4.** Behavioural patterns behind the demise of the commons are robust across nations. (*a*) Estimated parameter values show that while the virtual forest's state (parameter $\beta_R$) and the effort of others (parameter $\beta_{\langle T \rangle}^{-1}$) inform participant decisions, the Spanish participants exhibit a form of decision-making 'inertia' by which the current effort strongly reflects previous own efforts (parameters $\beta_T^{-1}$ to $\beta_T^{-5}$). The effect is significant up to five lags in the past. Here, shown are the parameter estimates (points) and the corresponding 95% confidence intervals (error bars). (*b*) Adjustments of the Spanish parameter values to fit the data from China indicate that the Chinese participants exhibit the same decision-making 'inertia' as their counterparts in Spain. The effect is only slightly weaker at lag 2 (parameter $\Delta\beta_T^{-2}$), but otherwise statistically indistinguishable between the two countries. The effort of others also has a statistically indistinguishable effect. The only qualitative difference is reflected in the $\Delta\beta_R$ parameter, revealing that the Chinese (respectively, Spanish) participants exert more effort when the resource is scarce (respectively, abundant). This is consistent with a gentler (respectively, steeper) initial decline of the resource in China (respectively, Spain). The negative relationship between resource abundance and effort in China backs up our conclusion from the time-series analysis (electronic supplementary material, table S2) that six additional Chinese groups would have eventually depleted the resource.

## 3. Discussion

Having asked participants from China and Spain to exploit a virtual forest while facing the same epistemic and socio-economic obstacles as real-world operators, we found that seemingly different outcomes are, in fact, remarkably similar and bode ill for the fate of common-pool resources. An exploratory data analysis in the form of clustering reveals that the results are largely attributable to three behavioural types (also called phenotypes in the literature), dubbed aggressive, moderate and timid. Although the nature of the game in our experiment is different from those in previous experiments that report behavioural phenotypes [36–38], we see clear parallels between aggressive, moderate and timid players herein and defectors, cooperators and supercooperators in [38], respectively. The consistency of previously identified behavioural phenotypes [37,38] further suggests that the types we found are also a consistent feature of human behaviour rather than a peculiarity of the specific experimental set-up. In fact, having worked with two geo-socially distant populations, and in a novel and relatively complex context, our results go a long way in fortifying the conclusions of the cited studies that human behaviours in social dilemmas are divisible into a small number of stable phenotypes.

A previous study [39] using a similar set-up, albeit with explicit resource 'dynamics' such that every 10 standing trees yielded one new tree per round, reported the outcome of the game experiment compared to other situations. Here, by contrast, we implemented a more realistic dynamic—whose qualitative characteristics, but not quantitative details, are known by the participants—and identified collective behavioural mechanisms that underpin decisions on exploitation, thus pointing to one main culprit for similarly dismal outcomes in both countries. Instead of prioritizing the resource state when deciding the current effort, participants operate under decision-making 'inertia' by which they are much more concerned with their own past efforts. A surprising aspect here is that this mechanism materializes in two populations not only separated geographically, but also influenced by a myriad of

confounding factors such as age, education and culture. The Chinese participants shared comparatively young age, exposure to higher education and upbringing in the midst of a quintessential East Asian cultural heritage. The Spanish participants mirrored the general population in terms of age and educational background, while socio-culturally belonging to a typical western democracy. Given that the same mechanism materialized despite these large differences, we concluded that behavioural patterns behind the demise of the commons are highly robust to confounding factors. It remains open for future research to explore if changes in the experimental design would yield significant differences between populations. For example, changing the profit per tree by adjusting the price of trees or the unit cost of effort would affect the strength of the underlying dilemma, and thus provoke more or less logging. Whether participants from different countries would be equally sensitive to variations in dilemma strength remains unclear at the moment.

Global environmental risks can no longer be contained without cooperation at an unprecedented scale in human history [40–42], but does humankind have what it takes to achieve such cooperativeness? The existence of collective behavioural patterns that are robust given a specific contextual situation is a reason for cautious optimism. In the case of common-pool resource exploitation, for example, encouraging a shift in focus from one's own past decisions to the resource state should reduce overexploitation in China and Spain alike. The aim here is to raise awareness of problematic behaviours, unlike experimental treatments that try to evoke a cooperative state of mind by indirect suggestion, e.g. by exploiting a known cognitive bias [24]. More generally, robustness promises that precautionary policies or educational programmes, when crafted with great care, may curb risky behaviours across continents and cultures. Pursuing this promise, therefore, has the potential to become an attractive research agenda for a wide variety of multidisciplinary studies on the origin of human cooperation.

# 4. Material and methods

## 4.1. Protocol and purpose

We devised an experimental protocol for the present study in October 2017. To implement this protocol, we conducted a total of four sessions of the experiment between November 2017 and March 2018. Two Spanish sessions took place on 29 November and 15 December 2017 at the Experimental Economics (Nectunt) Lab of the University of Zaragoza. Two additional sessions took place on 25 January and 8 March 2018 at the School of Computer Science and Technology of the Northwestern Polytechnical University, Xi'an, China.

The protocol envisioned that each session would involve multiple groups of six participants playing a common-pool resource game. We instructed all groups to exploit a virtual forest, and promised a monetary payout in RMB or EUR proportional to individual performance. By 'exploit', we meant a series of decisions by which participants select their preferred weekly logging effort, ranging 0–7 with decimal numbers permitted. An effort too high would bring in high profits early in the game, but would deplete the forest quickly thereafter. An effort too low would be sustainable, but would yield subpar profits due to underexploitation. Our purpose was to probe participant decision-making in the described context, for which we recruited 96 and 90 participants from China and Spain, respectively (electronic supplementary material, table S1). Over the course of 31 played games, we recorded a total of 9300 individual decisions that guaranteed a sufficiently large sample for the subsequent computational–statistical analyses.

## 4.2. Sessions of the experiment

Each session of the experiment consisted of three stages. In the preparatory stage, we directed participants to random seats, thus keeping the members of the same group physically separated. This was a precautionary measure in addition to never revealing who is playing with whom. We then instructed participants to read a detailed game tutorial, and confirm that they were ready to start playing by clicking on an *Accept* button at the end of this tutorial. The game would begin once six participants from the same group indicated their readiness.

In the gameplay stage, participants played 50 rounds of the game. The exact number of rounds remained undisclosed to avoid the final-round effects, i.e. a change in behaviour due to the impending end of the game. Importantly, we ensured that participants were *a priori* made aware of the unspecified number of rounds to prevent misdirection. To enable informed decision making, we prepared a computer interface

that graphically displayed the current state of the virtual forest, as well as (i) own effort and the average effort of other players from the preceding round, (ii) own number of cut trees and the average number of trees cut by others, also from the preceding round, and (iii) own profit and the average profit of others from the beginning of the game (electronic supplementary material, figure S1). Upon consulting all this information, participants were supposed to enter their decisions into a dedicated input form. If there were no inputs within a timeout period of 30 s, we assumed that the player is satisfied with their previous decision and the computer would repeat that same decision. Only in the first round, if necessary, the computer would make a random choice. In China and Spain, respectively, the timeout was reached in 14.3% and 4.5% of decisions.

In the final, payout stage, we paid participants a show-up fee and a supplement according to their performance in the game. The show-up fee equalled ¥40 in China and €5 in Spain. The supplement was constrained by a fixed budget available for each session of the experiment. Specifically, after subtracting the amount needed to cover all show-up fees, we divided the remaining budget to participants in proportion to their game profit. In China, this procedure yielded an average payout of ¥62.6 with a minimum of ¥27 and a maximum of ¥112. In Spain, the average payout was €15.1 with a minimum of €8 and a maximum of €23. At the time of writing, the exchange rate between the two currencies was ¥1≈€0.13.

## 4.3. Ethics statement

The experiment was approved by the Ethics Committee on the Use of Human Participants in Research of the Northwestern Polytechnical University and by the Clinical Research Ethical Committee of The Aragon Health Sciences Institute (IACS), and carried out in accordance with all relevant guidelines. We obtained informed consent from all participants. We furthermore ensured complete anonymity throughout all sessions of the experiment by avoiding any references to real names or demographic variables connecting participants to data.

## 4.4. Ecological and economic aspects

To emulate a virtual forest responsive to human decisions, we ran a background ecological model that evaluated tree regrowth against the posted logging efforts. We assumed that the forest is divided into $M$ patches, with each patch interpreted as the minimum space needed to grow a tree. Thus, the higher the number of empty patches, the higher is the number of growing trees. As patches become increasingly occupied and the carrying capacity is approached, the growth rate drops to zero. In mathematical terms, $dR/dt \propto g(M - R)$, where $R$ is the number of occupied patches, i.e. the forest's state, and $g$ is the maximum *per capita* growth rate with dimension time$^{-1}$.

To increase realism of the ecological model, we also incorporated an Allee effect, a phenomenon whereby population size correlates with the mean individual fitness of the population [35]. In particular, if the number of trees is very low, then reproductive success becomes highly unlikely because, e.g. trees are on average too far from one another for a pollinator to carry pollen. We implemented this by having the growth rate abruptly drop to zero when the number of trees is below the no-recovery threshold, i.e. when $R < R_c$. Using the Heaviside step function, $H(R - R_c)$, the resulting mathematical expression is $dR/dt \propto g(M - R)H(R - R_c)$. This function equals unity (respectively, zero) if its argument is positive (respectively, negative).

To convert decisions on logging effort into a number of cut trees, we assumed that there exists a characteristic time scale, $\tau$, to find and process a single tree. However, if the forest's state is low compared to the carrying capacity, searching takes longer and is extended by a factor of $M/R$. Accordingly, when the $i$th player inputs effort $T_i$, the corresponding number of cut trees is $C_i = T_i/(\tau M/R)$. The full dynamical equation that evaluates tree regrowth against logging efforts is

$$\frac{dR}{dt} = g(M - R)H(R - R_c) - \sum_i C_i. \tag{4.1}$$

We assessed the economics of logging in the virtual forest by calculating profits as a difference between revenues and costs. Revenues originate from selling the number of cut trees $C_i$ at constant price $p$. For the $i$th participant who decides to spend $T_i$ days a week logging, this yields a total weekly revenue of $pC_i = pT_iR/\tau M$. Apart from generating revenues, logging also incurs costs due to, e.g. fuel consumption or equipment wear and tear. Here, we assumed that a total weekly cost of logging is $cT_i$, where $c$ is the unit

cost of effort. This quantity is commonly used in ecological economics to express 'harvesting' costs on a per-person and per-unit-of-time basis [43]. A weekly profit, i.e. the difference between revenues and costs, finally becomes $\pi_i = pC_i - cT_i = T_i(pR/\tau M - c)$.

An optimal fair strategy should generate the same maximum profit for all participants. Such a strategy is achievable if everyone chooses the same optimal effort, $T^*$, by which the forest's state approaches an equilibrium, $R^*$. In this equilibrium, the total number of cut trees is $\sum_i C_i = NT^*R^*/\tau M$, where $N$ is the number of participants. Determining $T^*$ and $R^*$ involved maximizing profit $\pi^* = T^*(pR^*/\tau M - c)$ under the equilibrium condition, $g(M - R^*) = NT^*R^*/\tau M$. By equating the first derivative of profit with zero, we obtained $T^* = g\tau M/N[(p/\tau c)^{1/2} - 1]$ and $R^* = M(\tau c/p)^{1/2}$. With these quantities in hand, it was natural to classify those participants who posted efforts $T_i \leq T^*$ (respectively, $T_i > T^*$) as cooperators (respectively, defectors).

## 4.5. Data analyses

We analysed the data using two complementary techniques. Clustering, a form of unsupervised machine learning, helped us to independently identify predominant behavioural phenotypes among participants from China and Spain. A behavioural regression model subsequently discerned factors that largely explain how participants make decisions on harvesting effort. Both techniques are described in great detail in the electronic supplementary material, Methods.

Data accessibility. The datasets generated and analysed in the current study are available in the Open Science Framework repository, doi:10.17605/OSF.IO/AMG3C [44].

Authors' contributions. M.J., F.M.-C., C.G.-L. and Y.M. designed the research. F.M.-C., C.G.-L., C.L. and Z.W. performed the experiment. M.J. and F.M.-C. analysed data. All authors discussed the results and wrote the manuscript.

Competing interests. We declare we have no competing interest.

Funding. M.J. acknowledges support from the Japan Society for the Promotion of Science (grant no. 20H04288). Z.W. acknowledges support from the National Natural Science Foundation of China (grant no. U1836106). Y.M. acknowledges support from the Government of Aragón, Spain (grant no. E36-20R), Ministry of Economy and Competitiveness (MINECO) and European Fund for Regional Development (grant no. FIS2017-87519-P), the European Commission's Future and Emerging Technologies (FET) Proactive projects Dolfins (grant no. 640772) and Ibsen (grant no. 662725), the University of Zaragoza (Project no. 2015/022PIP) and from Intesa Sanpaolo Innovation Center. The funders had no role in study design, data collection and analysis, decision to publish or preparation of the manuscript.

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
