## [Reviewer comments · Royal Society Open Science]

Review History

RSOS-201026.R0 (Original submission)

Review form: Reviewer 1

Is the manuscript scientifically sound in its present form?

Yes

Are the interpretations and conclusions justified by the results?

Yes

Is the language acceptable?

Yes

Do you have any ethical concerns with this paper?

No

Have you any concerns about statistical analyses in this paper?

No

Recommendation?

Accept with minor revision (please list in comments)

Comments to the Author(s)

In "Behavioral patterns behind the demise of the commons across different cultures" authors improve our understanding of what it might take to govern the commons in a sustainable manner. They propose a social dilemma experiment in which participants gain profit from harvesting a virtual forest vulnerable to overexploitation. Research based on 16 Chinese and 15 Spanish groups shows that only one group from each country converged to the maximum sustainable yield, while other groups failed due to being overzealous and overexploitative. It is also shown that this may be due to the interplay between three prominent player behaviors, which are equally pervasive among players from both nations.

Understanding why we fail at social dilemmas is an intensely investigated subject with obvious practical ramifications. Human experiments, together with methods of network science and mathematical modeling, as well as statistical analysis, have been used successfully and with much effect recently to shed light on the problem from many different perspectives. They have also revealed different ways towards better understanding such behavior. In this sense, the study certainly addresses a relevant setup, and it also delivers results that will surely be of interest to the readership of Royal Society Open Science.

I have enjoyed reading this paper. I find it comprehensive and clearly written, and introducing new results that will surely also inspire future research along similar lines.

A revision should kindly address the following comments.

- 1) I didn't quite get Fig. 4A, in particular the outlier is rather strong, and the authors could do better here to explain or provide some rationale as to why.
- 2) In the introduction, I feel strongly that research on this subject from physics has been overlooked. The PGG game has been reviewed in *Statistical physics of human cooperation*, Phys. Rep. 687, 1-51 (2017). Common pool resource games have been studied in *Averting group failures in collective-risk social dilemmas*, EPL 99, 68003 (2012) and *Solving the collective-risk social dilemma with risky assets in well-mixed and structured populations*, Phys. Rev. E 90, 052823 (2014), for example.
- 3) Some references are weird for the context they suppose to cover, e.g. 15 for the public goods game. The point of reference 15 was to consider pool punishment, but as far as the game itself is concerned, there are many reviews as well as research papers that cover the PGG far more relevantly than ref. 15. The same for 13 for the resolution of social dilemmas. I suppose it is fine to cite one's own work even if it does not fit all that well, but then at least the authors should not make it look like nothing else has been done here and cite the relevant works of others as well, e.g. *Coevolutionary games - A mini review*, BioSystems 99, 109-125 (2010), where a lot has been covered in terms of what does and does not resolve social dilemmas.

Review form: Reviewer 2

Is the manuscript scientifically sound in its present form?

Yes

Are the interpretations and conclusions justified by the results?

Yes

Is the language acceptable?

Yes

Do you have any ethical concerns with this paper?

No

Have you any concerns about statistical analyses in this paper?

No

Recommendation?

Accept with minor revision (please list in comments)

Comments to the Author(s)

This work tackled an intriguing experiment of PGG, emulating a quite realistic woodcutter game, where more harvest given to each individual level and preserving the forest resource do compose a typical Common-type social dilemma structure, usually classified by Chicken-type dilemma.

The experiment was carefully designed, the obtained result as well as the discussion the authors highlighting seem quite conceivable and reliable.

Quite interestingly, as Fig. 2 showing, Chinese participants manage to keep forest's sustainability except for one group, while seven in Spanish subjects lost the control going down to the collapse level due to over-lumbering.

The approach the authors took in terms of experimental design, data handling, statistical analysis, seems quite robust.

Thus, I think this MS should be welcomed to the journal.

Yet, I would like to give some suggestions so as to enhance the significance and attractiveness to the audience as below.

What the authors presumed is what-is-called Lumber-jack Game, where not only plural players' actions but also the dynamics of environment whether it coming to recovered woods or depleted. With respect to the latter one, they presumed Eq. (4.1), which is quite acceptable. Such modeling there have been several previous works, one of the classical works relying on MAS approach should be cited;

Tanimoto; Environmental dilemma game to establish a sustainable society dealing with an emergent value system, *Physica D* 200, 1-24, 2005.

What-is-called dilemma strength, unlike 2 by2 explored by the previous work; cited by them as Ref [13], the present model cannot be quantified by simple parameters. One of the reasons is that the dynamics of environment abovementioned was defined as time-variable. But to some extent by means of experimental design, it can be controllable. I guess the social dilemma working behind this model would be increased if the reward per tree given to each participant is increased. If it is the case the result be different, thus Fig. 2 may show different picture. I wouldn't go as far as to say that the authors should do additional experiments at all. Yet I would like to hear from them what result would be expected to observe if such hypothetical settings are further explored.

Decision letter (RSOS-201026.R0)

Dear Dr Gracia-Lázaro,

On behalf of the Editors, I am pleased to inform you that your Manuscript RSOS-201026 entitled "Behavioral patterns behind the demise of the commons across different cultures" has been

accepted for publication in Royal Society Open Science subject to minor revision in accordance with the referee suggestions. Please find the referees' comments at the end of this email.

The reviewers and handling editors have recommended publication, but also suggest some minor revisions to your manuscript. Therefore, I invite you to respond to the comments and revise your manuscript.

- Ethics statement

- Data accessibility

If you wish to submit your supporting data or code to Dryad (<http://datadryad.org/>), or modify your current submission to dryad, please use the following link:
<http://datadryad.org/submit?journalID=RSOS&manu=RSOS-201026>

- Competing interests

- Authors' contributions

- Acknowledgements

- Funding statement

Because the schedule for publication is very tight, it is a condition of publication that you submit the revised version of your manuscript before 26-Jun-2020. Please note that the revision deadline will expire at 00.00am on this date. If you do not think you will be able to meet this date please let me know immediately.

Please note that Royal Society Open Science charge article processing charges for all new submissions that are accepted for publication. Charges will also apply to papers transferred to Royal Society Open Science from other Royal Society Publishing journals, as well as papers

submitted as part of our collaboration with the Royal Society of Chemistry (<https://royalsocietypublishing.org/rsos/chemistry>).

If your manuscript is newly submitted and subsequently accepted for publication, you will be asked to pay the article processing charge, unless you request a waiver and this is approved by Royal Society Publishing. You can find out more about the charges at <https://royalsocietypublishing.org/rsos/charges>. Should you have any queries, please contact openscience@royalsociety.org.

Kind regards,
Lianne Parkhouse
Editorial Coordinator
Royal Society Open Science
openscience@royalsociety.org

on behalf of Professor Matjaz Perc (Associate Editor) and Pete Smith (Subject Editor)
openscience@royalsociety.org

Reviewer comments to Author:

Reviewer: 1
Comments to the Author(s)

In "Behavioral patterns behind the demise of the commons across different cultures" authors improve our understanding of what it might take to govern the commons in a sustainable manner. They propose a social dilemma experiment in which participants gain profit from harvesting a virtual forest vulnerable to overexploitation. Research based on 16 Chinese and 15 Spanish groups shows that only one group from each country converged to the maximum sustainable yield, while other groups failed due to being overzealous and overexploitative. It is also shown that this may be due to the interplay between three prominent player behaviors, which are equally pervasive among players from both nations.

Understanding why we fail at social dilemmas is an intensely investigated subject with obvious practical ramifications. Human experiments, together with methods of network science and mathematical modeling, as well as statistical analysis, have been used successfully and with much effect recently to shed light on the problem from many different perspectives. They have also revealed different ways towards better understanding such behavior. In this sense, the study certainly addresses a relevant setup, and it also delivers results that will surely be of interest to the readership of Royal Society Open Science.

I have enjoyed reading this paper. I find it comprehensive and clearly written, and introducing new results that will surely also inspire future research along similar lines.

A revision should kindly address the following comments.

- 1) I didn't quite get Fig. 4A, in particular the outlier is rather strong, and the authors could do better here to explain or provide some rationale as to why.
- 2) In the introduction, I feel strongly that research on this subject from physics has been overlooked. The PGG game has been reviewed in *Statistical physics of human cooperation*, Phys. Rep. 687, 1-51 (2017). Common pool resource games have been studied in *Averting group failures*

in collective-risk social dilemmas, *EPL* 99, 68003 (2012) and Solving the collective-risk social dilemma with risky assets in well-mixed and structured populations, *Phys. Rev. E* 90, 052823 (2014), for example.

3) Some references are weird for the context they suppose to cover, e.g. 15 for the public goods game. The point of reference 15 was to consider pool punishment, but as far as the game itself is concerned, there are many reviews as well as research papers that cover the PGG far more relevantly than ref. 15. The same for 13 for the resolution of social dilemmas. I suppose it is fine to cite one's own work even if it does not fit all that well, but then at least the authors should not make it look like nothing else has been done here and cite the relevant works of others as well, e.g. *Coevolutionary games - A mini review*, *BioSystems* 99, 109-125 (2010), where a lot has been covered in terms of what does and does not resolve social dilemmas.

Reviewer: 2

Comments to the Author(s)

This work tackled an intriguing experiment of PGG, emulating a quite realistic woodcutter game, where more harvest given to each individual level and preserving the forest resource do compose a typical Common-type social dilemma structure, usually classified by Chicken-type dilemma. The experiment was carefully designed, the obtained result as well as the discussion the authors highlighting seem quite conceivable and reliable.

Quite interestingly, as Fig. 2 showing, Chinese participants manage to keep forest's sustainability except for one group, while seven in Spanish subjects lost the control going down to the collapse level due to over-lumbering.

The approach the authors took in terms of experimental design, data handling, statistical analysis, seems quite robust.

Thus, I think this MS should be welcomed to the journal.

Yet, I would like to give some suggestions so as to enhance the significance and attractiveness to the audience as below.

What the authors presumed is what-is-called Lumber-jack Game, where not only plural players' actions but also the dynamics of environment whether it coming to recovered woods or depleted. With respect to the latter one, they presumed Eq. (4.1), which is quite acceptable. Such modeling there have been several previous works, one of the classical works relying on MAS approach should be cited;

Tanimoto; Environmental dilemma game to establish a sustainable society dealing with an emergent value system, *Physica D* 200, 1-24, 2005.

What-is-called dilemma strength, unlike 2 by2 explored by the previous work; cited by them as Ref [13], the present model cannot be quantified by simple parameters. One of the reasons is that the dynamics of environment abovementioned was defined as time-variable. But to some extent by means of experimental design, it can be controllable. I guess the social dilemma working behind this model would be increased if the reward per tree given to each participant is increased. If it is the case the result be different, thus Fig. 2 may show different picture. I wouldn't go as far as to say that the authors should do additional experiments at all. Yet I would like to hear from them what result would be expected to observe if such hypothetical settings are further explored.

Author's Response to Decision Letter for (RSOS-201026.R0)

See Appendix A.

Decision letter (RSOS-201026.R1)

Dear Dr Gracia-Lázaro,

It is a pleasure to accept your manuscript entitled "Behavioral patterns behind the demise of the commons across different cultures" in its current form for publication in Royal Society Open Science. The comments of the reviewer(s) who reviewed your manuscript are included at the foot of this letter.

on behalf of Professor Matjaz Perc (Associate Editor) and Pete Smith (Subject Editor)
openscience@royalsociety.org

Associate Editor Comments to Author (Professor Matjaz Perc):

Comments to the Author:

Thank you for the comprehensive revision of your manuscript, which we are happy to accept for publication in Royal Society Open Science.

Appendix A

Zaragoza, June 21, 2020

Dear Editor,

Please find enclosed a revised version of our manuscript entitled “*Behavioral patterns behind the demise of the commons across different cultures*”, co-authored by Marko Jusup, Felipe Maciel Cardoso, Carlos Gracia-Lázaro, Chen Liu, Zhen Wang, and Yamir Moreno, which we are resubmitting for publication in *Royal Society Open Science*.

We are thankful to both reviewers for carefully reading our manuscript. We are enclosing herewith a detailed list of responses to all comments and suggestions raised by the reviewers. Having revised the manuscript as requested, we believe that it is now ready for publication.

Yours sincerely,

Dr. Carlos Gracia Lázaro, on behalf of all the authors.

Reviewer: 1

Comments to the Author(s)

In "Behavioral patterns behind the demise of the commons across different cultures" authors improve our understanding of what it might take to govern the commons in a sustainable manner. They propose a social dilemma experiment in which participants gain profit from harvesting a virtual forest vulnerable to overexploitation. Research based on 16 Chinese and 15 Spanish groups shows that only one group from each country converged to the maximum sustainable yield, while other groups failed due to being overzealous and overexploitative. It is also shown that this may be due to the interplay between three prominent player behaviors, which are equally pervasive among players from both nations.

Understanding why we fail at social dilemmas is an intensely investigated subject with obvious practical ramifications. Human experiments, together with methods of network science and mathematical modeling, as well as statistical analysis, have been used successfully and with much effect recently to shed light on the problem from many different perspectives. They have also revealed different ways towards better understanding such behavior. In this sense, the study certainly addresses a relevant setup, and it also delivers results that will surely be of interest to the readership of Royal Society Open Science.

I have enjoyed reading this paper. I find it comprehensive and clearly written, and introducing new results that will surely also inspire future research along similar lines.

Reply: We thank the reviewer for their effort and positive report. Please, find below our responses to the points raised.

A revision should kindly address the following comments.

1) I didn't quite get Fig. 4A, in particular the outlier is rather strong, and the authors could do better here to explain or provide some rationale as to why.

Reply: We thank the reviewer for pointing this out. In figure 4, we show the estimated values of the regression coefficients that appear in our statistical model of participant behaviour. The top panel (A) shows the estimates for the Spanish participants, while the bottom panel (B) shows how much the estimates for the Chinese participants deviate from those for the Spanish participants (i.e., from those in panel (A)). Four out of five

autoregressive coefficients, as well as the coefficient that characterises responsiveness to the effort of others are quantitatively the same for participants from both countries (deviations in panel (B) are statistically indistinguishable from zero). Where participants truly differ is their responsiveness to the resource state; the Chinese participants exert more effort when the resource is scarce, while the Spanish participants when it the resource is abundant. We made this clearer in the revised caption of figure 4.

2) In the introduction, I feel strongly that research on this subject from physics has been overlooked. The PGG game has been reviewed in Statistical physics of human cooperation, Phys. Rep. 687, 1-51 (2017). Common pool resource games have been studied in Averting group failures in collective-risk social dilemmas, EPL 99, 68003 (2012) and Solving the collective-risk social dilemma with risky assets in well-mixed and structured populations, Phys. Rev. E 90, 052823 (2014), for example.

Reply: We thank the reviewer for suggesting additional references. It would indeed seem that we have overlooked some relevant and helpful studies from physics. The revised manuscript cites them all.

3) Some references are weird for the context they suppose to cover, e.g. 15 for the public goods game. The point of reference 15 was to consider pool punishment, but as far as the game itself is concerned, there are many reviews as well as research papers that cover the PGG far more relevantly than ref. 15. The same for 13 for the resolution of social dilemmas. I suppose it is fine to cite one owns work even if it does not fit all that well, but then at least the authors should not make it look like nothing else has been done here and cite the relevant works of others as well, e.g. Coevolutionary games - A mini review, BioSystems 99, 109-125 (2010), where a lot has been covered in terms of what does and does not resolve social dilemmas.

Reply: We thank the reviewer for pointing out that more relevant work should have been referenced in this specific context. We corrected the mistake with the help of the reviewer's suggestions.

Reviewer: 2

Comments to the Author(s)

This work tackled an intrigued experiment of PGG, emulating a quite realistic woodcutter game, where more harvest given to each individual level and preserving the forest resource do compose a typical Common-type social dilemma structure, usually classified by Chicken-type dilemma.

The experiment was carefully designed, the obtained result as well as the discussion the authors highlighting seem quite conceivable and reliable.

Quite interestingly, as Fig. 2 showing, Chinese participants manage to keep forest's sustainability except for one group, while seven in Spanish subjects lost the control going down to the collapse level due to over-lumbering.

The approach the authors took in terms of experimental design, data handling, statistical analysis, seems quite robust.

Thus, I think this MS should be welcomed to the journal.

Reply: We thank the Reviewer for their valuable and positive report, which has contributed to improving our manuscript. Please, find below our responses to the points raised.

Yet, I would like to give some suggestions so as to enhance the significance and attractiveness to the audience as below.

What the authors presumed is what-is-called Lumber-jack Game, where not only plural players' actions but also the dynamics of environment whether it coming to recovered woods or depleted. With respect to the latter one, they presumed Eq. (4.1), which is quite acceptable. Such modeling there have been several previous works, one of the classical works relying on MAS approach should be cited;

Tanimoto; Environmental dilemma game to establish a sustainable society dealing with an emergent value system, Physica D 200, 1-24, 2005.

Reply: We thank the reviewer for suggesting this interesting study. We cite the study in the introduction when commenting about the resource dynamics.

What-is-called dilemma strength, unlike 2 by2 explored by the previous work; cited by them as Ref [13], the present model cannot be quantified by simple parameters. One of the reasons is that the dynamics of environment abovementioned was defined as time-variable. But to some extent by means of experimental design, it can be controllable. I guess the social dilemma working behind this model would be increased if the reward per tree given to each participant is increased. If it is the case the result be different, thus Fig. 2 may show different picture. I wouldn't go as far as to say that the authors should do additional experiments at all. Yet I would like to hear from them what result would be expected to observe if such hypothetical settings are further explored.

Reply: We thank the reviewer for this interesting remark. Indeed, the experimental design should have profound effects on participant behaviour and controllability of common pool resources. It is unclear to us in which direction these effects would manifest, but for sure they are an important subject to be explored in the future. Thus, we expanded the discussion section to briefly comment on this.